# Collective polarization dynamics in bacterial colonies signify the occurrence of distinct subpopulations

**Marc Hennes\*, Niklas Bender, Tom Cronenberg, Anton Welker, Berenike Maier** [ID]\*

Institute for Biological Physics, and Center for Molecular Medicine Cologne, University of Cologne, Cologne, Germany

\* marc.hennes@uni-koeln.de (MH); berenike.maier@uni-koeln.de (BM)

**Data Availability Statement:** All relevant data are within the paper and its Supporting Information files.

## Abstract

Membrane potential in bacterial systems has been shown to be dynamic and tightly related to survivability at the single-cell level. However, little is known about spatiotemporal patterns of membrane potential in bacterial colonies and biofilms. Here, we discovered a transition from uncorrelated to collective dynamics within colonies formed by the human pathogen *Neisseria gonorrhoeae*. In freshly assembled colonies, polarization is heterogeneous with instances of transient and uncorrelated hyper- or depolarization of individual cells. As colonies reach a critical size, the polarization behavior transitions to collective dynamics: A hyperpolarized shell forms at the center, travels radially outward, and halts several micrometers from the colony periphery. Once the shell has passed, we detect an influx of potassium correlated with depolarization. Transient hyperpolarization also demarks the transition from volume to surface growth. By combining simulations and the use of an alternative electron acceptor for the respiratory chain, we provide strong evidence that local oxygen gradients shape the collective polarization dynamics. Finally, we show that within the hyperpolarized shell, tolerance against aminoglycoside antibiotics increases. These findings highlight that the polarization pattern can signify the differentiation into distinct subpopulations with different growth rates and antibiotic tolerance.

## Introduction

Bacteria actively maintain a negative membrane potential as part of the ion motive force. Functioning as a source of energy, ion motive force powers ATP synthesis, transport across the membrane, and membrane-standing molecular machines [1–3]. Recent studies investigating membrane potential of *Escherichia coli* and *Bacillus subtilis* at the single-cell level revealed that the membrane potential is highly dynamic and heterogeneous [4,5]. Transient hyperpolarization is associated with increased death rate [6], while depolarization has been shown to maintain viability under oxygen depletion [7] and antibiotic treatment. In particular, aminoglycoside antibiotics induce hyperpolarization [6,8] and hyperpolarized cells tend to grow more slowly and have a higher death rate [6]. These reports provide evidence that

**Funding:** This work was supported by the Center for Molecular Medicine Cologne (www.cmmc-uni-koeln.de) through grant B6 and the Deutsche Forschungsgemeinschaft (www.dfg.de) through grant MA3898 granted to BM. The funders had no role in study design, data collection and analysis, decision to publish, or preparation of the manuscript.

**Competing interests:** The authors have declared that no competing interests exist.

**Abbreviations:** LHS, left-hand side; PCA, protocatechuic acid; PCD, protocatechuate-dioxygenase; ThT, thioflavin; TMRM, tetramethylrhodamine methyl ester; T4P, type 4 pilus.

inhibition of hyperpolarization maintains growth and viability. While there is no evidence of collective hyperpolarization in bacterial populations, spatially propagating waves of membrane depolarization have been found in *B. subtilis* colonies [9]. These waves coordinate the metabolic state and growth behavior between the interior and the periphery through the release of intracellular potassium via dedicated ion channels [9,10]. To our knowledge, collective dynamics of membrane potential in colonies and biofilms has not been found in other species so far.

Biofilms are an abundant form of bacterial life [11]. Within biofilms, localized gradients of nutrients, oxygen, and waste provide habitat diversity [12]. One consequence of this diversity is that bacteria partition into fast growing cells at the surface of the biofilm and slowly growing cells at the center [12,13]. Slow growth tends to increase tolerance of the bacteria against different antibiotics and other stresses [12,14]. Heterogeneity and dynamics of membrane potential potentially contribute to local habitat formation and antibiotic tolerance. Well-studied biofilm-formers like *Pseudomonas aeruginosa* initiate biofilm formation by surface attachment of single planktonic cells and subsequent proliferation into colonies [15,16]. The irreversible transition from the planktonic state into the biofilm state is believed to occur gradually at this stage [16]. In this study, we address these dynamics using *Neisseria gonorrhoeae* (gonococcus), the causative agent of the second most prominent sexually transmitted disease, gonorrhea [17]. By contrast to *P. aeruginosa*, *N. gonorrhoeae* uses type 4 pilus (T4P)-driven motility to self-aggregate into surface-attached spherical microcolonies consisting of thousands of cells within a few minutes [18–20]. T4P are extracellular polymers that continuously elongate and retract [21–24]. T4P dynamics are crucial for the structure of gonococcal colonies; a tug-of-war mechanism fluidizes the colonies, introducing local liquid-like order and causing colonies to form spheres [25–30]. Colony formation protects *N. gonorrhoeae* against the β-lactam ceftriaxone [31] and the degree of tolerance depends on the physical properties of the colonies [29]. Within several hours, a gradient of growth rates develops in these colonies [32], but it is unclear how and at which time scale habitat diversity emerges. Such transitions are expected to occur at some point in the maturation process of the biofilm and may be linked to cell differentiation and the emergence of sleeper cells [12,14]. The spherical geometry makes gonococcal colonies an ideal system for studying the evolution of the membrane potential during maturation of freshly assembled colonies into biofilms.

In this study, we focus on the dynamics of membrane potential in gonococcal colonies during early colony development. Using single-cell analysis within spherical colonies, we investigate the polarization dynamics at different stages of colony development. In freshly assembled colonies, single-cell membrane potential is heterogeneous and spatially uncorrelated. Eventually, a shell of hyperpolarized cells occurs at the colony center and travels towards the colony periphery. This event signifies a transition to collective membrane potential dynamics and correlates with reduction of growth. Behind the hyperpolarized shell, the intracellular potassium concentration increases and cells depolarize. A reaction–diffusion model strongly suggests that the dynamical pattern of oxygen concentration shapes the polarization dynamics. Application of the protein synthesis inhibitors kanamycin or azithromycin reverses the direction of the traveling shell. Within the shell, tolerance against kanamycin increases. Taken together, we show that membrane potential dynamics signifies the occurrence of habitats with different growth rates and antibiotic tolerance.

## In freshly assembled colonies, the membrane potential of neighboring cells is uncorrelated

Driven by type 4 pili, *N. gonorrhoeae* cells actively assemble into colonies comprising thousands of bacteria [18,29]. We characterized the membrane potential of single cells within

freshly assembled colonies growing in flow chambers that continuously provide fresh medium. The fluorescence intensity of the Nernstian dye TMRM was determined using confocal microscopy. Depending on the electrical potential $V_m$ across the inner membrane, the cationic dye partitions between the cytoplasm and the extracellular space [4]. The fluorescence intensities inside and outside of the cells were used to measure the membrane potential as described in the Materials and methods. Since the membrane potential is negative, an increase in fluorescence intensity indicates an increase in polarization, i.e., the potential becomes more negative.

We found that the TMRM signal in young colonies is highly heterogeneous (Fig 1A and Fig i in S1 Text) From the difference in TMRM fluorescence intensities outside of and within the cells, we derived the membrane potential $V_m$ of single cells. Potential pitfalls of this quantification are discussed in the Materials and methods. $V_m$ shows a Gaussian distribution (Fig 1B) with a mean value of $V_m = -95 \, mV$ which is slightly lower than the value obtained for single cells [33]. With time, the mean membrane potential of cells in the colony decreases by roughly 15 mV, indicating steady depolarization of the population (Fig 1B). The distribution of $V_m$ departs from the initial Gaussian shape towards a distribution with heavier tails at larger potentials, attributable to a randomly distributed subpopulation maintaining strong polarization (Fig 1A and 1B).

At the single-cell level, transient changes in polarization have been reported [4–6,8,34,35]. To find out whether the membrane potential is dynamic at the level of individual cells within colonies, we tracked the TMRM signal of single cells. To ensure that fluctuations in TMRM intensity are not caused by vertical movement of bacteria, we simultaneously tracked the sfGFP signal of the respective cells (strain *wt green*, Table i in S1 Text). A small fraction of cells experiences events of transient depolarization (Figs iiA and iiB in S1 Text) or hyperpolarization (Figs iiC and iiD in S1 Text and S1 Movie) that last for approximately 1 to 2 min. For all observed events, we find that transient changes in polarization (roughly 15 mV change) are uncorrelated between neighboring cells. To quantify correlations in changes of the membrane potential, we defined a measure of how strongly deviations from the mean TMRM fluorescence intensity, $I(t,x,y) - \langle I \rangle$, are spatially correlated between cells. $\langle I \rangle$ is the mean intensity of the colony at time t. In particular, we calculated the 2D polar correlation function $I_{corr}(t, \Delta\varphi, d) = (I(t, \varphi + \Delta\varphi, d) - \langle I \rangle)(I(t, \varphi, d) - \langle I \rangle)/\langle I \rangle^2$ at different depths $d$ of the colony ($d = 0 \, \mu$m at the edge). This function measures the correlation strength of the TMRM fluorescence signal in angular direction. We analysed $I_{corr}$ for 20 colonies and found little correlation (Fig 1C, further example in Fig iii in S1 Text). We note that gonococci tend to form diplococci whose membrane potential appears correlated over a time scale 1 h. This analysis confirms that the dynamics of transient changes in cellular membrane potential occurs independently between neighboring cells.

We conclude that in freshly assembled colonies, the membrane potential of single cells is heterogeneous and spatially uncorrelated. Individual cells transiently hyper- or depolarize at the time scale of minutes while most of the population gradually depolarizes at a time scale of hours.

## In large colonies, a shell of hyperpolarized cells travels through the colony and marks the transition from volume to surface growth

As the colonies grew in size, we found that the membrane potential transitioned from the uncorrelated dynamics described above to collective dynamics. Sometime after colony assembly in flow chambers, cells in the center of the colony showed increased polarization relative to the mean of the colony (Fig 2A and S2 Movie). Henceforth, we will refer to this relative increase in polarization as (collective) hyperpolarization. Collective hyperpolarization was

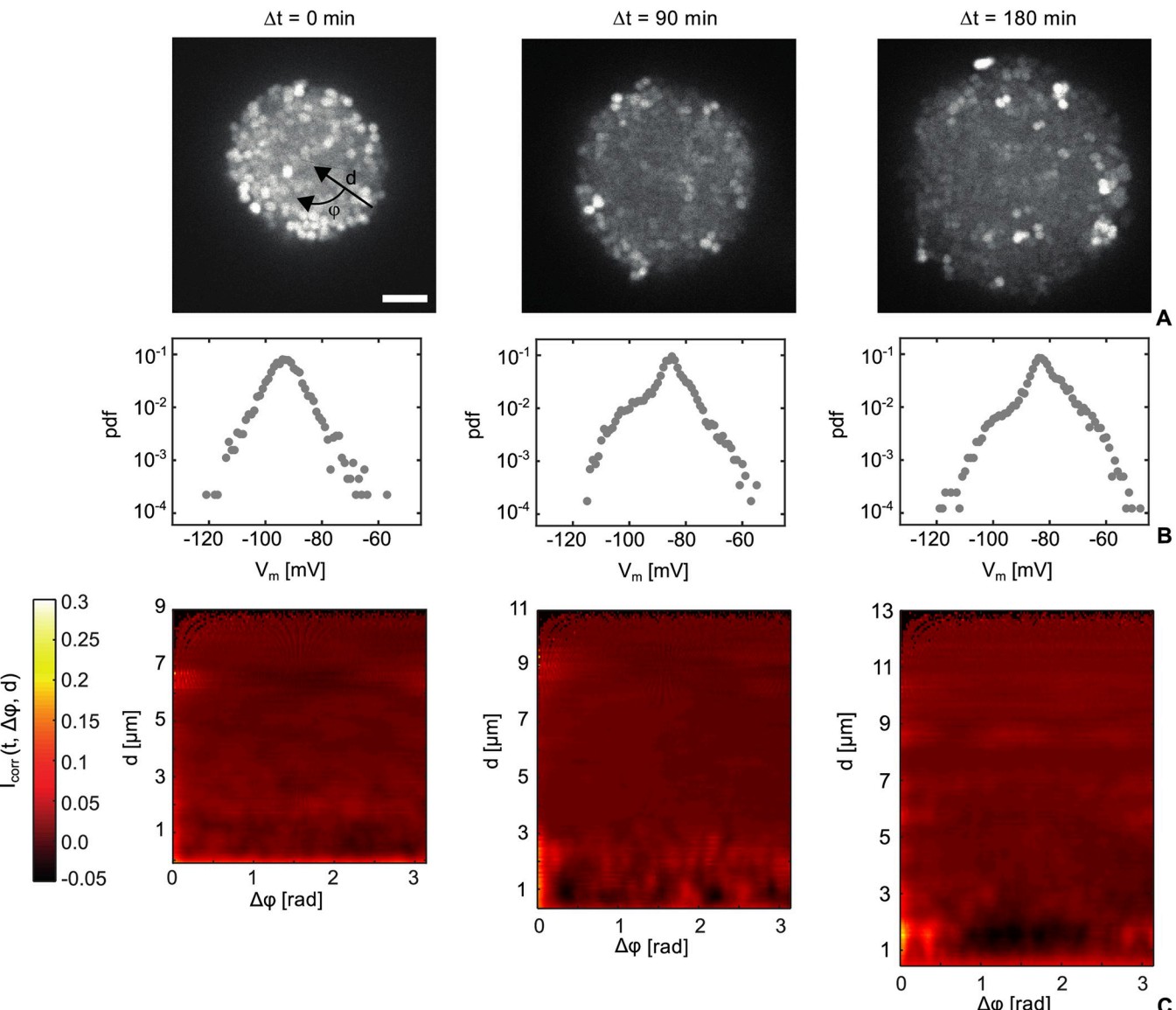

**Fig 1. Heterogeneity of membrane potential in small colonies formed at different time points.** Strain *wt green* (NG194), flow chamber. (A) Typical fluorescence intensities from confocal slices through colony center. Scale bar: 5 μm. (B) Probability density function (pdf) of membrane potential of single cells at Δt = 0, 90 min, 180 min. (>4,000 cells for each time point). (C) Angular correlation of the intensity fluctuations within the colony shown in (A) $I_{corr}(t, \Delta\varphi, d) = <\frac{(I(t, \varphi+\Delta\varphi, d) - <I>)(I(t, \varphi, d) - <I>)}{<I>^2}>_\varphi$ (color coded) as a function of the angular position difference Δφ and the distance d from the edge of the colony (see S1 Data for raw values).

transient and propagated radially from the center towards the edge of the colony across all cells. Notably, the shell of hyperpolarized cells rarely reached the edge and mostly became stationary a few *μm* away from the interface. The angular correlation coefficient shows a clear maximum at the position of the hyperpolarized shell (Fig 2B), indicating collective changes in membrane potential. The shell has a thickness of 3 to 4 μm, corresponding to 3 to 4 cell diameters. The difference of the membrane potential between the hyperpolarized cells and the cells at the colony center is $\Delta V_m \approx -10\ mV$ (Fig 2C).

Previously, we showed that within colonies of approximately 2 to 3 h of age, a characteristic pattern of growth rates forms whereby the rates decrease from the edge of the colony towards

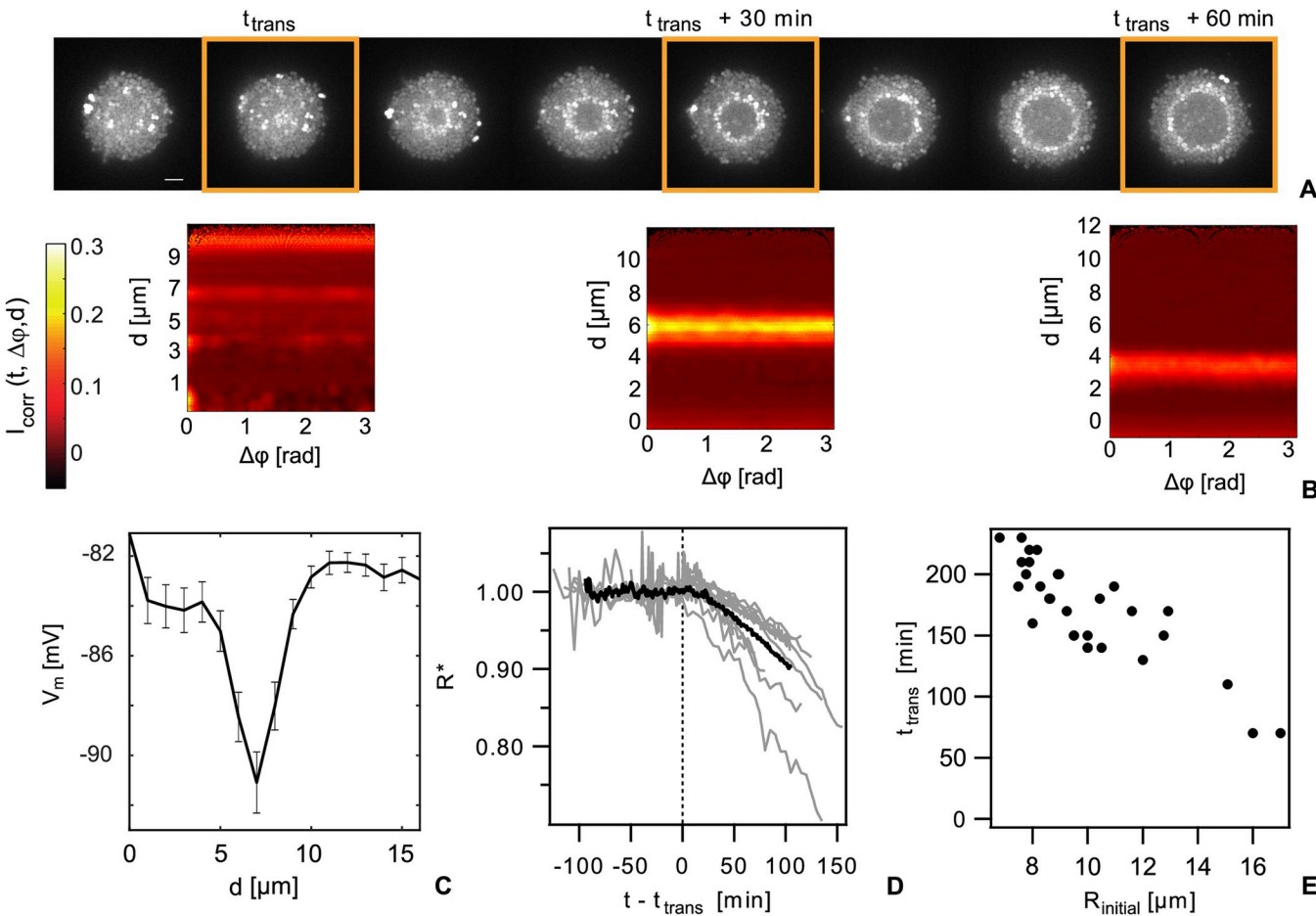

**Fig 2. Traveling shell of hyperpolarized cells in large colonies.** Strain *wt green* strain (NG194). $t_{trans}$ is the time point at which the membrane potential transitions to collective behavior. (A) Typical time lapse of TMRM fluorescence in flow cell, beginning 3 h after incubation. Scale bar: 5 μm. (B) Angular correlation of the intensity fluctuations as a function of radial position within the colony $I_{corr}(t, \Delta\varphi, d) = < \frac{(I(t,\varphi+\Delta\varphi,d)-<I>)(I(t,\varphi,d)-<I>)}{<I>^2} >_\varphi$. The images correspond to the fluorescence images in (A) at the time points indicated by orange frames. (C) Radial membrane potential of cells in colonies at the time point where the hyperpolarized cells reside 7 μm from the edge of the colony (*d*: distance from the edge of the colony). (D) Colony radius $R(t)$ normalized to an exponential function $R^*(t) = R(t)/\left(R_0 e^{\frac{\lambda_0 t}{3}}\right)$. The time axis was shifted to set t = 0 at the time when collective hyperpolarization occurs in the TMRM image. $\lambda_0$: initial growth rate, $R_0$: initial colony radius. Deviation from 1 indicates deviation from exponential growth. Black line: mean of 10 colonies from 3 different measurement days. Gray lines: individual colonies. (E) Time point of the transition to collective behavior as a function of initial colony size in static cultures (see S1 Data for raw values).

its center [32]. We assessed whether growth inhibition and the onset of collective membrane dynamics are correlated. Deviation from an initially exponential growth would indicate reduced growth rates within colonies. We determined the radii R of individual colonies as a function of time. As previously shown [32], the radius increases exponentially in freshly assembled colonies for approximately 2 h, i.e., $R(t) = R_0 e^{\frac{\lambda_0 t}{3}}$, with the initial radius $R_0$ and the growth rate $\lambda_0$. To detect the deviation from exponential growth, we normalized $R(t)$ by the radial growth function of the first 2 h, yielding $R^* = R(t)/(R_0 e^{\frac{\lambda_0 t}{3}})$. $R^* = 1$ signifies volume growth and $R^* < 1$ surface growth. For the individual colonies, we found that the deviation from exponential growth coincided with the onset of collective hyperpolarization (Fig 2D). Prior to the formation of the hyperpolarization shell ($t-t_{trans} < 0$), colony radii grew exponentially, $R^* \approx 1$, with an average growth rate $\lambda_0 \approx 0.6\ h^{-1}$ in line with previous measurements [32]. After shell

formation ($t-t_{trans} > 0$), the normalized radius $R^*$ quickly fell below 1. Relating the instantaneous radial position of the shell with the colony radius $R(t)$ indicates that growth cessation is limited to regions of the colony through which the hyperpolarized shell has already passed (Materials and methods, and Fig iv in S1 Text). Previously, we showed that the growth rate is slowest at the colony center while a narrow layer of cells continues to grow unperturbed [32]. The layer of growing cells [32] and the layer of cells that has not experienced transient hyperpolarization both comprise 3 to 4 cell diameters. We conclude that collective hyperpolarization in our gonococcal colonies coincides with growth arrest and marks the transition from volume to surface growth.

Within a single field of view in the microscope, the colonies did not transition simultaneously into the collective state (S3 Movie). Larger colonies transitioned earlier than smaller colonies. To characterize the relation between colony size and time point of the transition systematically, colonies were grown in static cultures without continuous flow, which accelerated the transition to collective polarization. We found that indeed, that the delay decreased with increasing initial size of the colony (Fig 2E). Furthermore, with increasing initial cell density, the delay decreased (Fig v in S1 Text), suggesting that collective hyperpolarization is caused by depletion of nutrients or oxygen.

In conclusion, we discovered a transition in membrane polarization dynamics from independent to collective changes in membrane potential that depends on colony size and marks the transition from volume to peripheral growth.

## In the wake of the hyperpolarized shell, intracellular $K^+$ levels increase and cells depolarize

After collective hyperpolarization has passed, cells depolarize slightly but significantly (Figs 2C and 3A) and transient changes in polarization as shown in Fig ii in S1 Text disappear. In previous reports, potassium flux has been shown to be tightly correlated with changes in membrane potential [7,9]. We addressed the question whether $K^+$ ions are involved in the membrane potential dynamics of gonococcal colonies. To this end, we applied the $K^+$- selective ionophore valinomycin to static cultures once the hyperpolarized shell had formed. Addition of valinomycin removed hyperpolarization and the TMRM fluorescence was homogeneously reduced throughout the colony (Fig iv in S1 Text). These results indicate that inhibition of the $K^+$ gradient abolishes the characteristic membrane potential pattern.

To investigate the dynamics of intracellular $K^+$, we stained $wt^*$ colonies with TMRM and a reporter for intracellular $K^+$ concentration, IPG-2 AM (Fig 3A–3C). We found that the IPG-2 AM fluorescence intensity strongly increased in the central region after the shell of hyperpolarized cells had passed, demonstrating that the concentration of potassium ions increased. This finding suggests that depolarization is at least partially caused by an influx of $K^+$ ions. We cannot exclude, however, that other ionic species are involved in the formation of the spatial polarization pattern.

In summary, after the hyperpolarization shell has passed, cells in its wake experience an influx of potassium ions and the membrane potential weakens.

## A reaction–diffusion model of oxygen uptake captures the dynamics of shell propagation

Next, we investigated the underlying mechanisms of shell formation and propagation. The onset of collective hyperpolarization depends on colony size (Fig 2E), strongly suggesting that it is related to a local gradient building up within the colony. The fact that the time point of collective polarization also depends on the total concentration of cells (Fig v in S1 Text) indicates

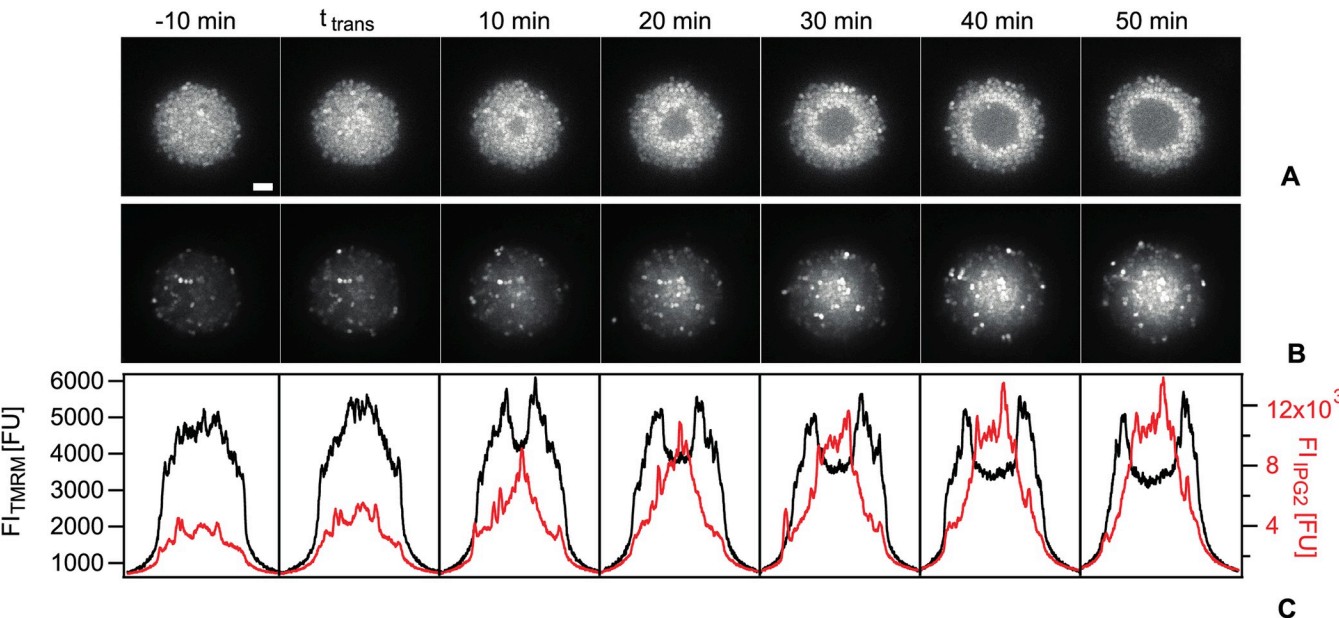

**Fig 3. Membrane depolarization correlates with the influx of K⁺ ions.** Strain *wt** (NG150), static culture. (A) Time lapse of TMRM fluorescence and (B) the K⁺ reporter IPG-2 AM across the equator of a colony. Scale bar: 5 μm. (C) Intensity profiles along the ROI shown in (A) and (B) (see S1 Data for raw values).

that depletion of a growth resource triggers collective hyperpolarization. The degree of oxygenation is thought to be an important factor that influences the chemistry and physiology of a microenvironment [13,14]. Because biofilm morphology shapes the oxygen concentration profile [36], we expect that oxygen deprivation is strongest in the center of the colony, where the hyperpolarization shell originates. Hypoxia has been shown to explain heterogeneous growth in biofilms using reaction–diffusion theory [37].

To assess whether oxygen gradient formation can explain the dynamics of the hyperpolarized shell, we first tested whether a 2D reaction–diffusion model (see Materials and methods) predicts the features of shell propagation on the observed time scales. In our model, the oxygen gradient changes with time because cells proliferate causing colony growth (Fig 4A). We assume that the hyperpolarized shell is triggered when the oxygen concentration $c(t,x,y)$ falls below a critical value $c^*$. Therefore, in our simulations, we tracked iso-concentration lines corresponding to the concentration $c^*$, which we assume to signify the position of the hyperpolarized shell propagation. To account for growth arrest, we set the proliferation rate of cells behind these circles to zero. Furthermore, we assume that the oxygen uptake rate is reduced in this area by a factor $\varepsilon$ (between 0 and 1).

We found that the propagation of iso-concentration circles qualitatively predicts the experimental dynamics of the hyperpolarization shell. Circles of concentration $c^*$ start at the center of the colony and propagate outward at close to constant speed (Fig 4A and 4B). As the circle approaches the edge of the colony, propagation captured through the relative position $r_{iso}/R$ slows down and stops several micrometers away from the boundary. The specific distance to the interface is governed by the free parameter $\varepsilon$; if cells at the colony center consume less oxygen, the circle propagates more slowly and equilibrium between oxygen diffusion and uptake is reached further inside the colony (Fig 4B). To compare the dynamics predicted by the model with the experiment, we characterized the propagation dynamics of the hyperpolarized shell by determining the normalized radius of the shell, $r_{hyper}/R$, as a function of time (Fig 4C). In agreement with the simulations, the shell propagated with a constant radial velocity early

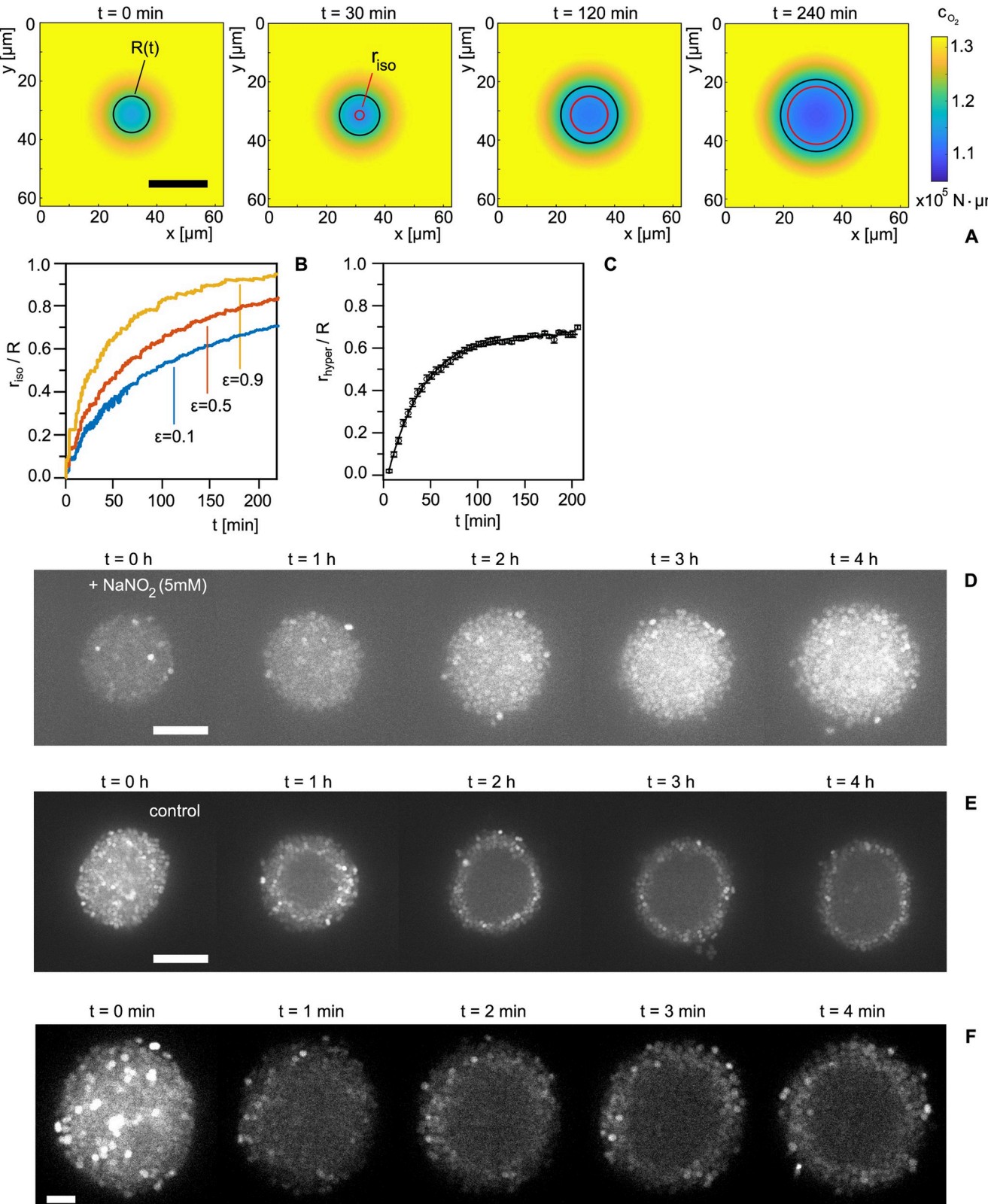

**Fig 4. Simulation of oxygen gradients and addition of alternative electron acceptors suggest that oxygen depletion triggers collective hyperpolarization.**
(A) Time lapse of simulated oxygen concentration field. Color code: oxygen concentration $c_{O_2}(t, x, y)$. Black circles: colony radius $R(t)$. Red circles: radial

distance $r_{iso}$ of the iso-concentration line at concentration $c^*$. Scale bar: 20 $\mu$m. (B) Simulated propagation of the $r_{iso}$ circle corresponding to $c^* = 0.8c_0$ (curves from simulations are shown for the uptake parameters $\varepsilon = 0.1, 0.5$, and $0.9$). (C) Experimental propagation of transient hyperpolarization shells. (Means and standard errors over 20 colonies.) (D) Time lapse of TMRM signal for a colony (strain $wt^*$, Ng150) in static culture supplemented with 5 mM of NaNO$_2$. Scale bar: 10 $\mu$m. (E) Time lapse of TMRM signal for a colony (strain $wt^*$) in static culture without supplement. Scale bar: 10 $\mu$m. (F) Time lapse of TMRM signal for a colony (strain $wt^*$) in static culture supplemented with the oxygen scavenging system PCA and PCD (see S1 Data for raw values). Scale bar: 5 $\mu$m. PCA, protocatechuic acid; PCD, protocatechuate-dioxygenase.

after the onset of hyperpolarization and subsequently slowed down as it reached the colony periphery. The early propagation velocity of $0.11 \pm 0.01$ $\mu m/min$ and the characteristic propagation time $\tau_{hyper} = 38 \pm 1$ $min$ obtained by an exponential fit to the data in Fig 4C, $r_{hyper}/R = A$ $(1-e^{-t/\tau})$, agree with the range of values obtained for the simulations (Fig 4B). We note, however, that not all parameters required for the model have been characterized experimentally for *N. gonorrhoeae* (see Materials and methods), and, therefore, we do not expect exact quantitative agreement between simulation and experiment.

If a concentration field of a growth resource is indeed involved in the onset and dynamics of the hyperpolarization shell, we expect the transiently hyperpolarized region to be broader and less defined for colonies with reduced cell number density and/or local ordering and non-spherical geometries. In this case, average radial gradients would be smeared out. Spherical colony shape and local order require the T4P retraction motor PilT [23,25,26,38] and, consequently, Δ*pilT* strains form non-spherical colonies. We repeated the flow chamber experiments with Δ*pilT* colonies and found that polarization dynamics of early colonies was comparable to $wt^*$ colonies. With time, the centers of colonies depolarized following the geometry of the aggregate, and, as predicted, there was only a weak and broad signature of collective hyperpolarization (Figs viiA and viiB in S1 Text). To find out whether this difference between $wt^*$ and mutant colonies was caused by lack of T4P retraction or change in colony architecture, experiments with the strain $pilT_{WB2}$ that has strongly reduced T4P retraction activity [25], but still forms spherical colonies with local liquid-like order [25] were carried out. $pilT_{WB2}$ colonies qualitatively behaved like $wt^*$ colonies (Figs viiC and viiD in S1 Text), indicating that colony morphology rather than T4P retraction was important for spatially correlated hyperpolarization.

Our model implies that the hyperpolarization shell propagates by transient hyperpolarization and not by flow of permanently hyperpolarized cells. In the following, we argue that the observed dynamics is qualitatively and quantitatively different from the radial mass flow caused by cellular proliferation. The growth-related velocity is minimal at the center of the colony and reaches its maximum at the edge of the colony [32] in contrast to the result shown in Fig 4C. The growth-related flow velocity is 4- to 5-fold lower compared to the propagation velocity of the hyperpolarized shell. At the same time, transient collective hyperpolarization is faster than the reshuffling of caged cells induced by pili retractions [27]. Therefore, we exclude cell flow of permanently hyperpolarized cells (either growth or pili-driven) as the cause of shell propagation.

The propagation dynamics of the hyperpolarized shell agrees well with the dynamics of iso-concentration rings of oxygen. At this point, however, we cannot exclude that other growth resources like glucose underlie the observed polarization dynamics. Oxygen acts as the final electron acceptor in the respiratory chain of the bacteria [39]. If oxygen triggers collective hyperpolarization, then addition of an alternative electron acceptor should retard or inhibit formation of the hyperpolarized shell. We investigated this point by supplementing the medium with sodium nitrite, which acts as an electron acceptor in the truncated denitrification pathway of *N. gonorrhoeae* [39–41]. We found that colonies supplemented with sodium

nitrite exhibit no transition to collective membrane potential dynamics over the time course of 4 h (Fig 4D). Conversely, control experiments consistently show collective polarization dynamics after 30 to 60 min (Fig 4E). Furthermore, we exposed colonies to the oxygen scavenger 3,4-protocatechuic acid (PCA) together with its enzyme protocatechuate-dioxygenase (PCD) [33,42] and observe a polarization pattern reminiscent of the late stages of shell propagation (Fig 4F).

In summary, we present strong evidence that the emergence of transient hyperpolarization and collective membrane potential dynamics is triggered by oxygen depletion along concentration gradients that form and travel outward towards the edge of the colony in radial direction.

## Tolerance to kanamycin but not to ceftriaxone is increased within the hyperpolarized shell

We investigated whether transitioning into the collective state affects the survivability of cells in different conditions. In particular, we addressed the emergence of antibiotic tolerance. At the single-cell level, it was shown that transient hyperpolarization correlates with increased death rate [6,8], suggesting that survivability decreases within the region enclosed by the shell of hyperpolarized cells. The state of polarization influences the bactericidal effect of aminoglycosides [8,43] and depolarization within the shell would protect the cells. Furthermore, reduced cell growth affects the killing efficiency of certain antibiotics including β-lactams [44], predicting that cells within the hyperpolarized shell are more tolerant to this class of antibiotics than the cells outside.

First, we studied the time evolution of the fraction of dead cells within untreated colonies using the membrane-impermeable DNA stain SytoX. Once the shell of hyperpolarized cells had passed, the fraction of dead cells $f$ increased (Fig 5A). In particular, $f$ inside the hyperpolarized shell was 3.5-fold higher than outside of the shell (Fig 5B). This finding is consistent with earlier studies showing that the death rate of individual cells that underwent transient hyperpolarization was higher than the death rate of cells without hyperpolarization [6,8].

Next, we addressed the questions how treatment with the aminoglycoside kanamycin affects membrane potential dynamics and how the pre-formed polarization pattern impacts kanamycin tolerance. Two-hour-old colonies were treated with kanamycin at 10-fold MIC for 30 min. After removal of the antibiotics, we characterized the dynamics of the membrane potential and the fraction of dead cells. Transient treatment with kanamycin caused a gradual increase in polarization throughout the colony, typically starting at the edge of the colony and moving inward (Fig 5C). Surprisingly, for colonies which had already formed a shell of hyperpolarized cells, kanamycin inhibited its radial progression and often the shell slowly traveled back towards the center of the colony, where it disappeared if it reached the center. Similar dynamics were observed under azithromycin treatment (Figs viiiA and viiiB in S1 Text). To find out whether tolerance increased in the wake of the hyperpolarized shell, we compared the regions of the colonies that resided inside the hyperpolarized shell during antibiotic treatment with the outside region. The fractions of dead cells were determined 3 to 4 h after treatment, because cell death is detectable with SytoX at a delay of several hours [29]. To account for the fractions of dead cells in the absence of antibiotics, we subtracted the respective values (Fig 5B), $f^*_{kan} = f_{kan} - f_{control}$. We found that $f^*_{kan}$ within the shell was 4-fold lower than outside (Fig 5C and 5D).

Finally, we treated the colonies with the β-lactam antibiotic ceftriaxone using the same protocol described for kanamycin. In contrast to the protein synthesis inhibitors described above, ceftriaxone did not affect the membrane potential (Fig 5E). After subtraction of the fraction of

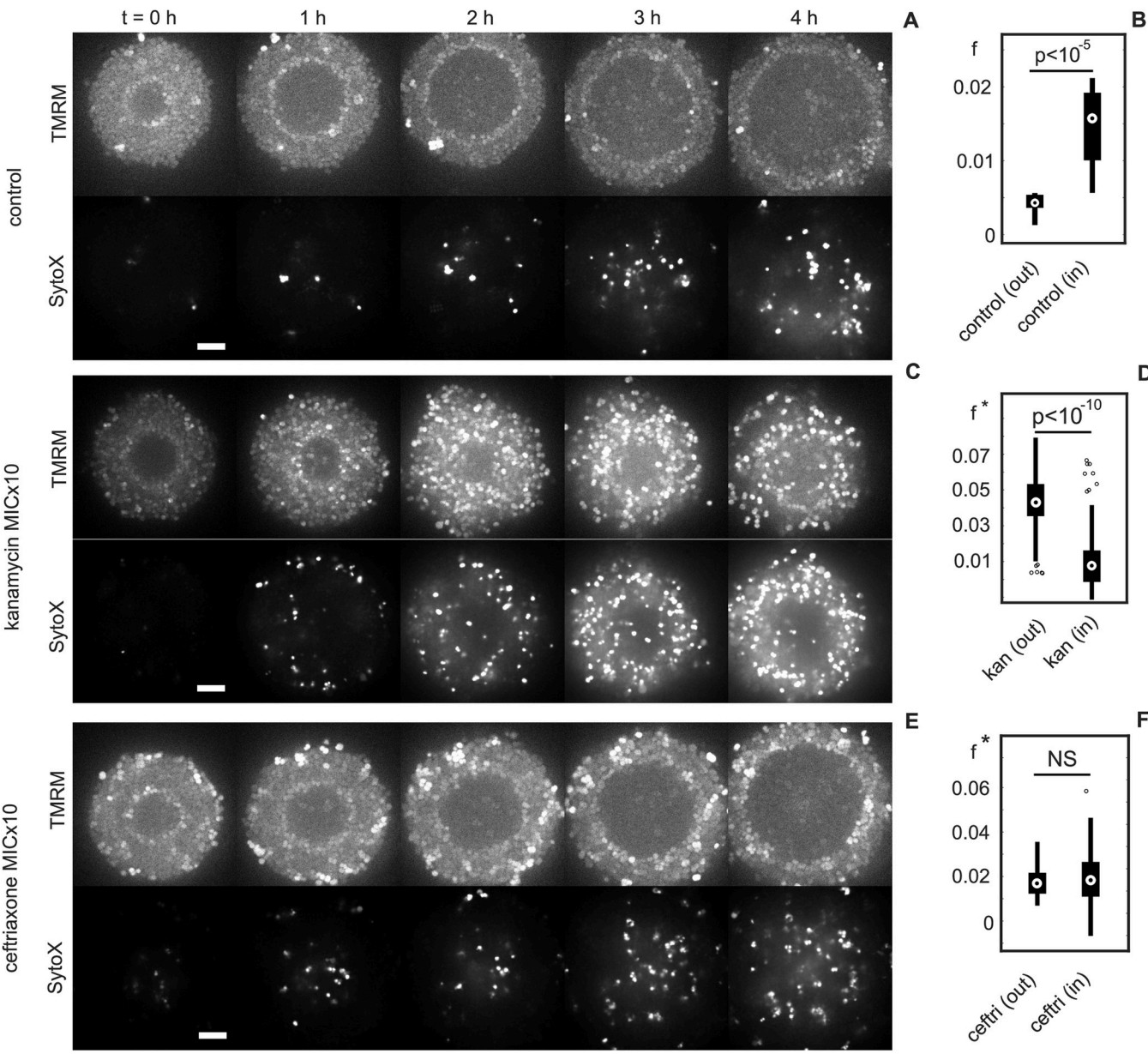

**Fig 5. Dynamics of membrane potential and fraction of dead cells *F* in colonies after 30 min of antibiotic treatment.** Strain *wt** (NG150), flow chamber. (A, C, E) Time lapse over 4 h of TMRM and SytoX fluorescence after respectively: no treatment, kanamycin treatment (MICx10, 200 $\mu g/ml$), and ceftriaxone treatment (MICx10, 0.04 $\mu g/ml$). Scale bar: 5 $\mu$m. (B, D, F) Fraction of dead cells *f* in the colonies 3–4 h later for both the region unaffected by the hyperpolarization shell (out) and the region affected (in) after respectively no treatment, kanamycin treatment, and ceftriaxone treatment. In the treated cases, $f^* = f - \langle f_{control} \rangle$. A total of 60 colonies evaluated for each condition. Bottom and top edges of the box indicate the 25th and 75th percentiles, respectively. Whiskers extend to the most extreme data points not considered outliers (marked as individual points). Central mark indicates the median of the distribution (see S1 Data for raw values).

dead cells without treatment, $f_{cef}^*$ was homogeneous throughout the colony (Fig 5E and 5F), showing that the pattern of membrane potential and the associated growth rate reduction at the center did not affect tolerance against ceftriaxone.

In summary, kanamycin treatment affects the polarization pattern of the colony, whereas ceftriaxone has no effect. Within the hyperpolarized sphere, cells were more tolerant against kanamycin but not against ceftriaxone.

## Discussion

Transient hyperpolarization has been reported at the single-cell level for *E. coli* and *B. subtilis* [4,6,8]. Importantly, hyperpolarization was uncorrelated between neighboring cells in these studies. Similarly, the membrane potential dynamics of freshly assembled gonococcal colonies studied here shows no nearest neighbor correlation. This suggests that gonococci, while being part of multicellular colonies, still behave like planktonic cells in the first few hours after assembly. By contrast to previous reports, we reveal a transition from independent to correlated hyperpolarization occurring once the colony is large enough.

Our results are most consistent with the following scenario. Governed by T4P motility and T4P-mediated cell-to-cell adhesion, gonococci assemble into colonies by T4P activity within minutes [23]. During the early stage of this colony, gonococcal behavior remains uncorrelated and colony formation is fully reversible [19]. Within the colony, a gradient of oxygen builds up. When a critical concentration of oxygen is reached, cells hyperpolarize. Once the oxygen concentration falls below this critical concentration, cells depolarize, are growth-inhibited, and consume less oxygen. Since the colony continues to grow at the surface, the critical iso-concentration shell (signified experimentally as a hyperpolarized shell) propagates towards the periphery of the colony. Depending on the oxygen consumption rates and the exogenous oxygen supply concentration, the shell of hyperpolarized cells becomes stationary at a specific distance away from the edge of the colony. An analytical solution of our reaction–diffusion model (see Materials and methods) predicts that this distance is related to the critical $O_2$ concentration $c^*$, the concentration at the surface of the colony, $c_0$, and the concentration $c_c$ at the center of the colony, $r_{iso}/R = \sqrt{(c^* - c_c)/(c_0 - c_c)}$. In the flow chamber, we find $r_{hyper}/R \approx 0.7$, which entails that the critical concentration is in the range $0.5c_0 < c^* < c_0$. Under static growth conditions, we observed that the hyperpolarized shell is closer to the edge of the colony (see, e.g., Fig 4E), in agreement with the $O_2$ concentration at the surface being lower than under continuous flow. It is also interesting to note that treatment with the protein synthesis inhibitors kanamycin and azithromycin halts or reverses the propagation of the hyperpolarized shell. As the antibiotics inhibit the growth of the colony and potentially further reduce the consumption rate of oxygen, our model predicts this observation. We note that our investigation is restricted to early stages of biofilm formation. In theoretical studies and experiments with other species, it has been shown that reaction–diffusion processes affect the architecture of biofilms [45,46]. In future studies, it will be interesting to find out whether reaction–diffusion models also predict the architecture of mature gonococcal biofilms. It is tempting to speculate that the effect of oxygen is more complex in this species as strong depletion induces colony disassembly [19].

While, we can clearly demonstrate that collective and transient hyperpolarization signifies the transition of volume to surface growth and changes in the state of antibiotic tolerance, the molecular mechanism of transient hyperpolarization remains unclear. According to our model, the $O_2$ concentration range that causes hyperpolarization is narrow. Hyperpolarization correlated with growth arrest or increased mortality has been reported in various studies [6–8]. One study identifies a cause for hyperpolarization; under aminoglycoside treatment, the inhibition of ribosomes frees ATP that causes the ATP synthase to work in ATP hydrolysis direction, increasing the proton gradient [8]. In our system, we rather expect the ATP levels to be reduced as a consequence of reduced proton gradient under low oxygen conditions. It will be very interesting to investigate the mechanism of transient hyperpolarization in future work.

We expected that reduced polarization and growth rate made gonococci more tolerant against aminoglycosides and β-lactams, respectively. The absolute decrease in polarization between the outside and the inside of the hyperpolarized shell amounts to a few millivolts only. Yet, we clearly observed that the fraction of dead cells is lower within than outside of the

shell after transient kanamycin treatment, indicating that cells are immediately protected against this aminoglycoside. Recent work shows that killing activity of aminoglycosides arises from dysregulated membrane potential [8]. They found that in depolarized cells, aminoglycosides exert a bacteriostatic effect. For our work, this strongly suggests that depolarization within the hyperpolarized shell causes tolerance against kanamycin. Yet, we cannot exclude that growth arrest also affects tolerance. By contrast, growth arrest within the hyperpolarized shell had no influence on tolerance against the β-lactam ceftriaxone. A previous study showed that tolerance quantitatively correlates with growth rate [44]. In that study, growth rates were varied by different but constant growth environments. The protective effect of growth inhibition may not occur instantaneously after reduction of growth rate explaining why gonococci were not protected against ceftriaxone despite growth inhibition.

In summary, we reveal a transition from uncorrelated single-cell fluctuations to collective dynamics of membrane polarization in gonococcal colonies and provide strong evidence that the local oxygen gradient governs this transition. Further, we find that a traveling shell of hyperpolarized cells demarks the differentiation into 2 subpopulations with different growth rates and different levels of antibiotic tolerance. In future studies, it will be important to find the underlying mechanism of transient hyperpolarization. We anticipate that studying dynamical polarization patterns in developing biofilms will provide novel insights into the mechanisms of biofilm differentiation in various bacterial species.

## Materials and methods

### Growth media and bacterial strains

Media composition, assay preparation, and setup follow [27]. Gonococcal base agar was made from 10 g/L dehydrated agar (BD Biosciences, Bedford, Massachusetts, United States of America), 5 g/L NaCl (Roth, Darmstadt, Germany), 4 g/L K2HPO4 (Roth), 1 g/L KH2PO4 (Roth), 15 g/L Proteose Peptone No. 3 (BD Biosciences), 0.5 g/L soluble starch (Sigma-Aldrich, St. Louis, MO), and supplemented with 1% IsoVitaleX (IVX): 1 g/L D-glucose (Roth), 0.1 g/L L-glutamine (Roth), 0.289 g/L L-cysteine-HCL × H2O (Roth), 1 mg/L thiamine pyrophosphate (Sigma-Aldrich), 0.2 mg/LFe(NO3) 3 (Sigma-Aldrich), 0.03 mg/L thiamine HCl (Roth), 0.13 mg/L 4-aminobenzoic acid (Sigma-Aldrich), 2.5 mg/L β-nicotinamide adenine dinucleotide (Roth), and 0.1 mg/L vitamin B12 (Sigma-Aldrich). GC medium is identical to the base agar composition but lacks agar and starch. Employed bacterial strains (Table i in S1 Text) are derivatives of strain *N. gonorrhoeae* MS11.

### Culture preparation for experiments in the flow chamber and under static conditions

For the microscopy experiments, we used 2 different setups. First, bacteria were grown in flow chambers. This condition ensures continuous flow of medium and approximately constant supply of growth resources. Second, bacteria were grown in static culture. This condition was used when fast depletion of growth resources was desired or supplements were added during the experiment. All experiments follow the same preparation protocol: Overnight cultures of the respective strain were suspended in fresh GC + IVX medium at $OD_{600}$ 0.1 unless noted otherwise. Under addition of 1% v/v ddH2O, the obtained bacterial solution was then incubated in a shaker at 37°C and 5% $CO_2$ for 30 min to 1 h, which allowed colonies to first disassemble and then reform freshly.

Flow chamber experiments: Cells were loaded into a microfluidic flow chamber (Ibidi Luer 0.8 mm channel height + Ibitreat) that was previously coated with Poly-L-Lysine (Sigma, Cat.

No. P4832, 50 μg/mL) overnight. The flow chamber was continuously supplemented with medium at a temperature of 37°C using a peristaltic pump (model 205U; Watson Marlow, Falmouth, United Kingdom) operating at 2 rpm. To visualize the membrane potential, TMRM (Sigma-Aldrich) was added to the supplied medium at a concentration of 0.1 $\mu$m [33].

Static condition experiments: 300 $\mu L$ of the bacterial suspension at an OD of 0.1 were pipetted into a $\mu$-slide 8 Well (Ibidi + Ibitreat), which was coated with Poly-L-Lysine (Sigma, Cat. No. P4832, 50 μg/mL) over the previous night. Together with the bacterial suspension, and depending on the experiment, we added 3 $\mu$m valinomycin (Sigma-Aldrich), 5 mM sodium nitrite $NaNO_2$ (Roth), or 20 $\mu g$/mL IPG2-AM (Ion BioSciences). Oxygen scavenger system (protocatechuic acid, PCA, and protocatechuate 3,4-dioxygenase, PCD, both from Sigma-Aldrich) preparation and concentration was taken from [42].

## Confocal imaging

For imaging, colonies of various sizes and diameters were randomly chosen and imaged using an inverted microscope (Ti-E Nikon) equipped with a CSU-X1 (Yokogawa) spinning disk unit, a 100× CFI Apo Tirf objective (Nikon), an EMCCD camera (iXon 897 X-11662, Andor), and 488 nm and 561 nm excitation lasers. If not stated otherwise, colonies were imaged at their equator once every minute (acquisition time 100 ms) in the brightfield, green, and red channel (laser power 3% and 2%, respectively).

## 3D movie of hyperpolarized shell progression

3D movies were obtained by recording multiple colonies inside the flow chamber every 5 min (strain NG194, *wt green*, acquisition time: 50 ms) for a total time interval of 2 h. For every imaging step, we recorded a z-stack of 10 μm with a z-spacing of 0.2 μm between each image close to the bottom of the microscope coverslip. Approximately 488 nm and 561 nm excitation lasers were set to 5% and 10% intensity, respectively. In addition, we acquired a brightfield image for every time point. Finally, 3D volume renderings and movies of hyperpolarization shell formation within colonies were created with NIS-Elements Ar Ver. 5.02 (Build 1271, Nikon).

## Characterization of antibiotic tolerance and fraction of dead cells

Colonies (*wt**, NG150) were incubated for approximately 120 min to allow the formation of the hyperpolarized shell within a subset of colonies. Subsequently, colonies were treated with medium containing kanamycin, ceftriaxone, or azithromycin at a concentration corresponding to 10× the MIC for kanamycin and ceftriaxone (200 $\mu g$/mL and 0.04 $\mu g$/mL, respectively) and 100× the MIC for azithromycin (0.64 $\mu g$/mL) for 30 min. Under control conditions, colonies were treated with medium that was supplemented with an equal volume of $ddH_2O$ instead. After the treatment was stopped, colonies were again supplemented with medium lacking any antibiotics but containing SytoX (Invitrogen) at a concentration of 0.2 $\mu$m. During this process, we acquired a z-stack of each colony every 10 min. The stack covered an area of 5 μm around the equatorial plane of the colonies, with a z-spacing of 0.5 μm between individual slices. Imaging was continued for 4 h beyond the treatment phase. In addition, brightfield images were acquired. The fraction of dead cells inside of colonies was then determined by tracking the green SytoX signal (Trackmate).

## Determination of single-cell potential

The cationic dye TMRM partitions between the cytoplasm (concentration $c_{in}$) and the extracellular space (concentration $c_{out}$) depending on the electrical potential $V_m$ across the inner

membrane [4]:

$$V_m = \frac{RT}{zF} \ln\left(\frac{c_{out}}{c_{in}}\right) \tag{1}$$

Here, $R$ is the gas constant, $T$ is the absolute temperature, $F$ is the Faraday constant, and $z$ is the valence of the dye ($z = +1$). Concentrations $c_{in,out}$ are proportional to the TMRM fluorescence intensities inside of cells and outside of the colonies, respectively. After loading, the TMRM signal equilibrates within approximately 5 min [33]. To derive $V_m$ from the fluorescence intensities, we proceeded as follows.

From the recordings, single-cell membrane potentials were obtained by first finding cell positions for each frame through the sfGFP signal using the Fiji plugin Trackmate. We then averaged the TMRM signal (red channel) over the area of the cells at these positions. Since TMRM molecules partition between the inside and outside of the cell depending on the presence of negative charges, we additionally measured the TMRM signal far away from the colonies. For both, we subtracted the contribution of the camera noise, which we quantified from images in the absence of cells and TMRM. Membrane potential values were then calculated according to Eq (1), with $T = 310K$.

We note that in this study, we have not corrected for effects of diffraction. In a previous study at the single-cell level, we did account for this effect and measured slightly (approximately 10%) more negative membrane potential [33]. The difference may be accounted to the fact that we did not correct for diffraction here, or to the fact that cells depolarize directly after colony formation. Importantly, the difference is small and does not affect the major conclusions of this study.

A recent study showed that the cationic Nernstian dye thioflavin T (ThT) can affect the growth rate of *E. coli* [47]. We measured the growth rate of individual gonococcal colonies in the presence of TMRM and found $\lambda = (0.5$–$0.6)h^{-1}$. This is comparable to the rate of colonies grown under the same conditions but in the absence of TMRM [32], suggesting that TMRM does not significantly affect the membrane potential. Moreover, the study with *E. coli* showed that an efflux pump impacts the ThT fluorescence signal [47]. We cannot exclude that efflux pumps affect the intracellular concentration of TMRM within gonococci, resulting in underestimated absolute values of the membrane potential. However, this systematic error would not affect the conclusions of this work.

## Localized growth rate in front of and behind the hyperpolarization shell

To show that growth rates decline after passage of the hyperpolarized shell, but not prior to it, we devised a simplified binary colony growth model where we assume that the cell growth rate $\lambda = \lambda_0$ is constant prior to the passage of the shell and $\lambda = 0$ afterwards. We start from the differential equation for the number of cells $N_r$ inside spherical shells at a distance $r$ from the center of mass of the spherical colony: $\dot{N}_r = \lambda(r, t)N_r$. With $N_r = \phi V_r$, where $\phi$ is the cell number density of the shell and $V_r$ its volume, we can write the temporal change in total colony volume as $\dot{V} = \int_0^R \lambda r^2 dr$ ($R$: colony radius). Using $\dot{V} = 4\pi\dot{R}R^2$, and splitting the integral in a part behind ($r < r_{hyper}$) and before ($r > r_{hyper}$) the hyperpolarized shell, we obtain a relation between colony radius $R$ and the instantaneous radial shell position $r_{hyper}$

$$R^2\left(R - \frac{3\dot{R}}{\lambda_0}\right) = r_{hyper}^3.$$

Before the hyperpolarized shell forms ($t < t_{trans}$), we expect the left-hand side (LHS) to be equal to zero. Afterwards, the LHS evolves as the third power of $r_{hyper}$. This relation is in good agreement with experimentally obtained values of $R$, $r_{hyper}$, (Fig iv in S1 Text, 15 colonies evaluated). Additionally, heterogeneous growth rates in the colonies several hours after incubation are in agreement with previous findings [32].

### Reaction–diffusion model for shell propagation dynamics

We numerically solved a 2D reaction–diffusion model for the concentration field of oxygen [45]

$$\partial_t c = -\kappa(c)\varrho + D\Delta c.$$

Here, $c := c(t,x,y)$ is the local concentration of $O_2$ inside and outside of the colony ($r = \sqrt{x^2 + y^2}$: distance to colony center-of-mass), $\kappa(c)$ is the oxygen consumption rate by each cell, $\varrho$ is the local cell number density, and $D$ is the diffusion coefficient of oxygen in the solvent. To account for both uptake- and diffusion-limited regimes, we used the Michaelis–Menten form $\kappa(c) = \kappa_0 \frac{c}{c+K_M}$, where $K_M$ is the Michaelis–Menten constant of the oxygen uptake reaction, and $\kappa_0$ is the upper limit of oxygen uptake per cell. As fixed parameters, we used the Michaelis constant determined for *Bacillus licheniformis*, $K_M = 2.5 \times 10^3$ *molecules* $\cdot$ $\mu m^{-3}$ [48], $c_{solvent} \equiv c_0 = 1.3 \times 10^5$ *molecules* $\cdot$ $\mu m^{-3}$ [33]. In bulk solvent, the diffusion coefficient of oxygen is $D_{O_2} = 1.5 \times 10^5 \mu m^2 \cdot min^{-1}$ [49]. To account for hindered diffusion in the colony, we set the oxygen diffusivity inside the colony to 1/10th this value. Finally, the product of the maximal oxygen consumption rate and the cellular density in colonies is $\kappa_0\varrho = 45 \times 10^5$ *molecules* $\cdot$ $\mu m^{-3}$ $min^{-1}$ [25,33]. The cell density was assumed to be constant inside a circle of radius $R$, where $R$ is the radius of the colony, and zero outside. To mimic nutrient supply in the flow chamber, we set $c = c_0$ for $r = R(t) + 5$ $\mu m$. We solved the model using a 2D DuFort–Frankel scheme in Matlab on a grid of size $128 \times 128$ in simulation units, corresponding to a window of $64 \times 64$ $\mu m^2$ ($ds = 0.5$ $\mu m$), with $c(t = 0,x,y) := c_0$ (timestep $\Delta t = 10^{-3}$ min). The initial colony radius was 6 $\mu m$. After $10^4$ steps (10 min), we obtained a steady-state concentration profile $c(t,x,y) := c(r) \equiv c_{eq}(r)$. At that moment, we allowed cells to proliferate with growth rate $\lambda = 0.01$ $min^{-1}$ [32] for 300 min and chose an adequate concentration $c^* = 0.8 \times c_0$, slightly smaller than $c_{eq}(0)$. From that moment, in regions where the concentration was lower, the growth rate was set to zero. The colony radius evolved as $\dot{R}(t) = \frac{\lambda}{3}\left(R(t) - \frac{r^3(t,c^*)}{R^2(t)}\right)$, where $r(t,c^*)$ is the distance of the iso-concentration circle to the colony center-of-mass. Additionally, we allowed the oxygen uptake rate in the region $r < r(t,c^*)$ to diminish by a factor $\varepsilon \in [0,1]$. To test the robustness of our system and results, we varied several key parameters (including oxygen diffusivity, distance of boundary condition) and found that variations had little to no qualitative impact on the results.

An analytical steady-state solution to the reaction–diffusion equation in terms of a Maclaurin series was derived in [50]. To quadratic order in the center-of-mass distance $r$, the solution reads $c(r) = c_c + \frac{\kappa_0\varrho c_c r^2}{3!D(c_c+K_M)}$, where $c_c$ is the (time-dependent) oxygen concentration at the center of the colony. From there, we obtain the relation $\frac{r_{iso}}{R} = \left[\frac{c^*-c_c}{c_0-c_c}\right]^{0.5}$ under the assumption that $c_0 \approx c(R)$. Because $c_c$ is larger than zero and cannot exceed the boundary value $c_0$, the critical concentration $c^*$ has range $0.5c_0 < c^* < c_0$.

## Supporting information

**S1 Text. Figures i–viii and Table i.**
(PDF)

**S1 Movie. Typical example of colony with blinking cell.** Movie corresponds to Fig iiA in S1 Text. Δt = 1 min.
(AVI)

**S2 Movie. Three-dimensional reconstruction of formation and propagation of the hyperpolarized shell.** Δt = 5 min.
(AVI)

**S3 Movie. Typical example of onset of collective hyperpolarization for colonies with different initial sizes.** Δt = 5 min.
(AVI)

**S1 Data. Raw values underlying the figures.**
(XLSX)

## Acknowledgments

We thank the Maier lab for stimulating discussions.

## Author Contributions

**Conceptualization:** Marc Hennes, Berenike Maier.

**Formal analysis:** Marc Hennes.

**Funding acquisition:** Berenike Maier.

**Investigation:** Marc Hennes, Niklas Bender, Tom Cronenberg, Anton Welker.

**Methodology:** Marc Hennes, Niklas Bender, Tom Cronenberg, Anton Welker.

**Project administration:** Berenike Maier.

**Software:** Marc Hennes.

**Supervision:** Berenike Maier.

**Validation:** Marc Hennes.

**Visualization:** Marc Hennes, Niklas Bender, Berenike Maier.

**Writing – original draft:** Marc Hennes, Berenike Maier.

**Writing – review & editing:** Marc Hennes, Berenike Maier.

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
