## [Editor Report · Decision Letter 0]

5 Sep 2022

Dear Dr. Maier, 

Thank you for submitting your manuscript entitled "Collective polarization dynamics in bacterial colonies" for consideration as a Research Article by PLOS Biology.

Your manuscript has now been evaluated by the PLOS Biology editorial staff, as well as by an academic editor with relevant expertise, and I am writing to let you know that we would like to send your submission out for external peer review.

Once your full submission is complete, your paper will undergo a series of checks in preparation for peer review. After your manuscript has passed the checks it will be sent out for review. To provide the metadata for your submission, please Login to Editorial Manager (https://www.editorialmanager.com/pbiology) within two working days, i.e. by Sep 07 2022 11:59PM.

Kind regards,

Paula

---

Senior Editor

PLOS Biology

---

## [Decision Letter · Decision Letter 1]

8 Nov 2022

Dear Dr. Maier,

Thank you for your patience while your manuscript "Collective polarization dynamics in bacterial colonies" went through peer-review at PLOS Biology. Your manuscript has now been evaluated by the PLOS Biology editors, an Academic Editor with relevant expertise, and by several independent reviewers.

In light of the reviews, which you will find at the end of this email, we are pleased to offer you the opportunity to address the comments from the reviewers in a revision that we anticipate should not take you very long. We will then assess your revised manuscript and your response to the reviewers' comments with our Academic Editor aiming to avoid further rounds of peer-review, although might need to consult with the reviewers, depending on the nature of the revisions.

Please address all the reviewers' issues. In particular, it is important that you improve the discussion of the technical details and biological interpretation of your findings.

**IMPORTANT - SUBMITTING YOUR REVISION**

*Resubmission Checklist*

*Published Peer Review*

*PLOS Data Policy*

*Blot and Gel Data Policy*

Sincerely,

Paula

---

Senior Editor

PLOS Biology

REVIEWS:

Reviewer #1: Membrane voltage in living systems.

Reviewer #2: Biofilms.

Reviewer #3: Modeling of bacterial systems, colony growth.

Reviewer #1: The manuscript by Hennes et al. investigates the role of membrane potential during aggregation of individual cells of N. gonorrhoeae. The authors identified a striking behavior or hyperpolarization in colonies that correlated with a change in growth behavior of the colony. The hyperpolarization was simulated which suggested oxygen depletion as a critical initation event for the hyperpolarization which was supported by experiments. Finally, they show a heterogeneous response to cell survival depending on the state of polarization in the colony. Overall, I believe the authors found a very interesting behavior which was highly reproducible with important physiological consequences. The paper was well written and clearly presented. I have only a few issues to contribute to this excellent manuscript.

1. The identification of K+ as the responsible ion. I'm not sure the authors can definitively say that K+ is ion dictating the hyperpolarization. When using valinomycin, it acts as a voltage clamp by enabling an ion specific flow. So it will set the voltage of all the cells to whatever is the Nerstian concentration difference between inside and outside the cells. The data helps confirm that it is indeed voltage hyperpolarization, but does not necessarily suggest K+. For example, changes in Na+ could be occurring, but those changes would just be offset by a flow in K+ resulting in the clamped cell. Second, the experiments with the potassium indicator were confusing. In many conventional cells (neurons, cardiomyocytes, E. coli), the potassium concentration inside the cell is high relative to the outside, such that it sets up a negative polarization. If one increases the cytoplasmic concentration of calcium, that would result in an increase in voltage (hyperpolarization). What I don't know is the concentration of K+ inside N. gonorrhoeae. If it is much lower than the medium concentration, it is possible that increased K+ could depolarize. A second alternative could be that the pH inside the cells is changing after reaching a critical O2 level. The crown-ethers are particulary sensitive to pH. If the authors want to make a stronger claim about the importance of K+, they should do the genetics (as in the Suel Nature paper), or use a fluorescent pH indicator to show that pH is not responsible for the APG-2 fluorescence changes. However, I believe the authors could change the language and cage their findings a bit without hurting the impact of the manuscript, too.

2. Small issue: After seceral reads, the data such as in figure 3c looked like time traces to me. I thought that the colonies repeatedly went high and then low as the x-axis is labeled in minutes. Can you separate the traces to make it clear that you are looking at a spatial profile for 1 time point, not a continuous time trace?

3. I would bring the data in S6 (oxygen depletion) to the main figure. The appearance of the shell at the outer edge is visually striking and I think it's worth elevating it to give it more impact.

4. For the claims of no changes in extracellular ions, is there data that could be shown in the supplement?

Reviewer #2: In this manuscript Hennes et al. explore the spatial dynamics of growth and membrane hyperpolarization/depolarization that occur in aggregated colonies of Neisseria gonorrhoeae (Nh). Using confocal microscopy, cellular resolution analysis, and a variety of fluorescent dyes they demonstrates that membrane polarization patterns are disordered in aggregates of Nh right after they have assembled via T4P activity. Shortly thereafter, they observe a spherical wave of hyperpolarization that proceeds from the colony center and travels toward the outer periphery, stopping several cell lengths from cell-liquid interface. Behind this wave, cells depolarize slightly and cell growth arrests. They show that the depolarization region behind the traveling shell corresponds to influx of potassium into the cells, and they use a reaction-diffusion model to support the interpretation that this polarization wave occurs together with the decrease in local oxygen concentration threshold required for cell growth and division. Supplementation with an alternative electron acceptor eliminates the formation of the polarization wave, at least on the time scale of these experiments. The growth-arrested region of the cell groups in their interior also corresponds to local tolerance to kanamycin. 

Overall I found this to be a very strong paper with good technical execution throughout, and explained very clearly in the text. It is VERY much appreciated that the authors *did not* succumb to the temptation to characterize this pattern of colony activity as a kind of analogue to neural activity, or embryo development, as other groups have recently done for Bacillus subtilis - in my opinion, with no real justification on logical or empirical grounds and at significant expense to the clarity of the biofilm research community. 

I have a few minor comments about a few points in the text, and a broader comment about some of the phrasing in the introduction and discussion that the authors may wish to consider. Beyond this I think this paper will be well received in the community and is a good fit to the general readership of this journal. 

1) The figure panels are not labelled very clearly - there are little letters in the bottom right of the multiple panel set to which the letters correspond. At first I did not see them, which made the figures a bit harder to read than they need to be. I would switch the letters to larger bolded versions in the top-left of each corresponding panel. 

2) Lines 187-189: it is noted that extracellular concentrations of K+ and N1+ were measured, as were intracellular concentrations of Ca2+ . . . but there is no reference to main text or SI figures in this sentence.

3) Lines 198-200: the paper states here that local oxygen concentration is a primary factor for the chemistry and physiology of a microenvironment. This is certainly true in many cases, but not universally true, e.g. in anaerobic environments and for bacteria that are not obligate aerobes like Nh (though it can use nitrite as an alternative e- acceptor). I would just change the wording here slightly so as not to overgeneralize. 

4) Lines 307-309: Can it be said for certain that the depolarization increases tolerance against kanamycin, or could it just be growth arrest, for which depolarization is a physiological indicator?

5) Line 336: Suggest changing to "Depending on the oxygen consumption rates and the exogenous oxygen supply concentration …" To this point, there is a substantial literature from the biofilm simulation field on the reaction-diffusion processes that govern the depth of actively growing layers in biofilms. These may be good to cite here. See in particular: PMID: 10099266

6) There is some language in the introduction and conclusion that makes it a bit ambiguous as to whether the authors think that groups of Nh cells are generating these polarization/depolarization "on purpose"; that is, whether they evolved to do this, or whether it is just a biophysical consequence of growing in a group with a limiting solute diffusing from the exterior and being depleted in the interior. The simplest explanation in my view would be the latter view; the cells do not have a regulatory switch from disordered to collective behavior, rather this transition occurs as a consistent outcome of the cell group geometry, cell nutrient/oxygen consumption, and nutrient/oxygen supply conditions. The majority of the language in the main text seems to imply this view, but there is text in the introduction and discussion that sometimes seems to imply that this is all part of an active, regulated differentiation process, rather than a byproduct of cell group growth. This was the only real point where the clarity of the test faltered somewhat to my reading, so I would advise taking a look out for this in a revision. 

Reviewer #3: This manuscript describes a previously unknown collective behaviour in developing colonies of the bacterium Neisseria gonorrorhea: a transition from uncorrelated, random polarization fluctuations of individual cells, to directional waves of hyperpolarization. The authors show convincingly that this phenomenon is triggered by oxygen depletion within the growing colony and is connected with growth arrest. This discovery is intriguing and potentially important since it may have wider relevance for biofilm development. The manuscript is well written and the findings are convincingly supported by data. I have a number of comments: 

1. The manuscript's conclusions rely heavily on the use of the Nernstian dye TMRM to measure membrane potential. The relationship between the readout of Nernstian dyes and membrane potential is not always simple, so that careful checking and calibration is needed (see, e.g. Mancini et al, (2020) Biophysical Journal 118, 4-14). In this manuscript the conclusions are mainly qualitative but it would still be reassuring to include some discussion on the validity of the dye for quantiative measurements.

2. Prior to the hyperpolarization wave, it is asserted that the membrane potential shows no spatial correlation (eg line 102 "a randomly distributed subpopulation maintaining strong polarization"). However, Figure 2b does seem to show some spatial structuring of the fluorescence, with brighter cells preferentially located at specific radii. It would be good to comment on this.

3. Is this random polarization and depolarisation that is seen at early times also seen in planktonic cells? I would have liked to see some discussion of whether it is a feature of early colony formation or a remnant of planktonic behavior.

4. Lines 143-6 "We determined the radii of individual colonies as a function of time (start radius 0 and growth rate 0) and normalized it by the radial growth function of the first 2 hours…" for me this needed more explanation. An additional sentence should be added to explain where the expression for the radial growth function in the first 2 hours comes from.

5. Lines 157-8 It is stated that the wave of hyperpolarisation precedes growth arrest. But the evidence that hyperpolarization comes first then growth arrest (not the other way around) could be made clearer. Perhaps this point could be strengthened?

6. Lines 299-301 It is intriguing that treatment with kanamycin apparently reverses the direction of the wave of hyperpolarization. The authors suggest in the discussion that this could happen because kanamycin prevents growth and hence depletion of oxygen. Perhaps this hypothesis could be tested using the already developed simulation model for oxygen dynamics.

---

## [Editor Report · Decision Letter 2]

8 Dec 2022

Dear Dr. Maier,

Thank you for your patience while we considered your revised manuscript "Collective polarization dynamics in bacterial colonies" for publication as a Research Article at PLOS Biology. This revised version of your manuscript has been evaluated by the PLOS Biology editors and the Academic Editor.

Based on our Academic Editor's assessment of your revision, we are likely to accept this manuscript for publication, provided you satisfactorily address the following data and other policy-related requests.

1. DATA POLICY:

A) Supplementary files (e.g., excel). Please ensure that all data files are uploaded as 'Supporting Information' and are invariably referred to (in the manuscript, figure legends, and the Description field when uploading your files) using the following format verbatim: S1 Data, S2 Data, etc. Multiple panels of a single or even several figures can be included as multiple sheets in one excel file that is saved using exactly the following convention: S1_Data.xlsx (using an underscore).

B) Deposition in a publicly available repository. Please also provide the accession code or a reviewer link so that we may view your data before publication.

Regardless of the method selected, please ensure that you provide the individual numerical values that underlie the summary data displayed in the following figure panels as they are essential for readers to assess your analysis and to reproduce it: Figures 1BC, 2BCDE, 3C, 4BC, 5BDF, and supplementary figures S2BD, S3, S4, S5, S6, S7BD.

We note that you have already provided the data for the majority of the figures.** Please also ensure that figure legends in your manuscript include information on where the underlying data can be found, **and ensure your supplemental data file/s has a legend.

2. We suggest a modification in the title to make it more declarative: "Collective polarization dynamics in bacterial colonies define differentiation into distinct subpopulations with different growth rates and antibiotic tolerance".

We expect to receive your revised manuscript within two weeks.

*Published Peer Review History*

*Press*

Sincerely,

Paula

---

Senior Editor,

pjaureguionieva@plos.org,

PLOS Biology

---

## [Editor Report · Decision Letter 3]

15 Dec 2022

Dear Dr. Maier,

Thank you for the submission of your revised Research Article "Collective polarization dynamics in bacterial colonies signify the occurrence of distinct subpopulations" for publication in PLOS Biology. On behalf of my colleagues and the Academic Editor, Victor Sourjik, I am pleased to say that we can in principle accept your manuscript for publication, provided you address any remaining formatting and reporting issues. These will be detailed in an email you should receive within 2-3 business days from our colleagues in the journal operations team; no action is required from you until then. Please note that we will not be able to formally accept your manuscript and schedule it for publication until you have completed any requested changes.

PRESS

Sincerely, 

Paula 

---

Senior Editor

PLOS Biology
